# Retroactive modulation of spike timing-dependent plasticity by dopamine

Zuzanna Brzosko, Wolfram Schultz, Ole Paulsen*

Department of Physiology, Development and Neuroscience, Physiological Laboratory, University of Cambridge, Cambridge, United Kingdom

**Abstract** Most reinforcement learning models assume that the reward signal arrives after the activity that led to the reward, placing constraints on the possible underlying cellular mechanisms. Here we show that dopamine, a positive reinforcement signal, can retroactively convert hippocampal timing-dependent synaptic depression into potentiation. This effect requires functional NMDA receptors and is mediated in part through the activation of the cAMP/PKA cascade. Collectively, our results support the idea that reward-related signaling can act on a pre-established synaptic eligibility trace, thereby associating specific experiences with behaviorally distant, rewarding outcomes. This finding identifies a biologically plausible mechanism for solving the 'distal reward problem'.

## Introduction

Spike timing-dependent plasticity (STDP) is a physiologically relevant form of Hebbian learning (*Caporale and Dan, 2008*). In its classic form, STDP depends on the order and precise timing of presynaptic and postsynaptic spikes: pre-before-post spike pairings induce timing-dependent long-term potentiation (t-LTP), whereas post-before-pre pairings induce timing-dependent long-term depression (t-LTD) (*Markram et al., 1997*; *Bi and Poo, 1998*). However, the quantitative rules of STDP are profoundly influenced by neuromodulators (*Seol et al., 2007*; *Pawlak et al., 2010*), including dopamine (DA) (*Zhang et al., 2009*; *Edelmann and Lessmann, 2011*; *Yang and Dani, 2014*). Although it is well established that reward-motivated behavior depends on the activity of DA neurons (*Schultz et al., 1997*; *Suri and Schultz, 1999*; *Pan et al., 2005*), the mechanisms that associate specific experiences with rewarding outcomes, which typically occur after a delay, are not well understood. This is referred to as the *distal reward problem* (*Hull, 1943*). To address this problem, here, we examined whether DA modulates STDP not only when applied during, but also—more importantly—when applied *after* the pairing event.

## Results and discussion

We first sought to corroborate the shape of the STDP induction curve by varying the time interval between the presynaptic and postsynaptic activity ($\Delta t$; *Figure 1A,B,G*). To this end, we monitored excitatory postsynaptic potentials (EPSPs) that were evoked by extracellular stimulation of the Schaffer-collateral-CA1 pathway during whole-cell recordings of CA1 pyramidal cells in mouse horizontal slices (postnatal days 12–18; 'Materials and methods'). Plasticity was induced in current clamp mode using an induction protocol that involved 100 pairings of a single EPSP followed by a single postsynaptic spike (t-LTP; *Figure 1A*) or a single postsynaptic spike followed by a single EPSP (t-LTD; *Figure 1B*) at 0.2 Hz. Consistent with previous studies (*Bi and Poo, 1998*; *Zhang et al., 2009*; *Edelmann and Lessmann, 2011*), the pre-before-post pairing protocol with $\Delta t = +10$ ms induced t-LTP (182 ± 14%; $t(4) = 5.6$, p = 0.0049 vs 100%, $n = 5$; *Figure 1A*) and the post-before-pre pairing protocol with $\Delta t = -20$ ms induced t-LTD (61 ± 9%; $t(4) = 4.4$, p = 0.0121 vs 100%, $n = 5$; *Figure 1B*). Surprisingly, however, we found that the post-before-pre pairing protocol with $\Delta t = -10$ ms instead of

*For correspondence: op210@ cam.ac.uk

**eLife digest** To help someone learn a new task, we might give them a reward after they have performed well. However, these rewards tend to be given several seconds or minutes after the behavior they are supposed to promote. Therefore, it is unclear how the rewards affect the brain and help accelerate the learning process.

Information is processed and sent around the brain by networks of cells called neurons. These networks are constantly remodeled because learning changes the connections—called synapses— that neighboring neurons signal across. Synapses can be strengthened so that signals are sent across them more easily in the future. Synapses can also be weakened, making it harder for the neurons to subsequently communicate.

A chemical called dopamine is often produced in the brain when a reward is received. If dopamine is present in a synapse whilst a neuron is signaling to its neighbor, it can affect how effectively this communication occurs. Brzosko et al. have now investigated whether dopamine can also change the synapses if it is applied after signaling has already happened.

The strengthening or weakening of synapses can be triggered by electrically stimulating the neurons on either side of a synapse at particular times. Brzosko et al. did this to neurons in slices of mouse brain, and then applied dopamine to the neurons. The results suggest that dopamine can reverse synaptic weakening and can even cause the synapses to strengthen. However, the dopamine had to be applied immediately after stimulation to be able to strengthen the synapse. The next challenge is to establish whether this change in synaptic strength is responsible for the change in behavior.

inducing t-LTD, elicited robust t-LTP (202 ± 21%; $t(6)$ = 5.0, p = 0.0025 vs 100%, $n$ = 7; *Figure 1C,D,G*). This conflicts with previous reports from hippocampal cultures (*Bi and Poo, 1998*; *Zhang et al., 2009*) and acute slices (*Edelmann and Lessmann, 2011*; *Yang and Dani, 2014*), where post-before-pre pairing protocols never elicited synaptic potentiation in baseline conditions. Given that DA has been found to widen the time window for the induction of t-LTP (*Zhang et al., 2009*; *Yang and Dani, 2014*), we wanted to assess whether endogenous DA could be responsible for the potentiation observed with the post-before-pre pairing under our experimental conditions. Therefore, we repeated this set of experiments using the post-before-pre pairing protocol with $\Delta t$ = −10 ms in the presence of DA receptor (DAR) antagonists. Indeed, combined application of the D1/D5 receptor antagonist SCH23390 (10 µM) and D2-like receptor antagonist sulpiride (50 µM) from the start of the recordings prevented t-LTP and enabled t-LTD instead (72 ± 8%; $t(5)$ = 3.6, p = 0.0160 vs 100%, $n$ = 6; *Figure 1C,D,G*), rendering the STDP induction curve similar to that observed in hippocampal cultures (*Bi and Poo, 1998*; *Zhang et al., 2009*). These results suggest a modulatory action of endogenous DA, presumably released during the pairing protocol (*Frey et al., 1990*; *Yang and Dani, 2014*), which resulted in the changed polarity of plasticity at narrow negative spike-timing intervals.

Next, we wanted to examine whether the application of exogenous DA during pairing at negative spike-timing intervals facilitates synaptic potentiation. Indeed, the post-before-pre pairing protocol with $\Delta t$ = −20 ms, which elicited robust t-LTD in control condition (74 ± 9%, $t(8)$ = 3.0, p = 0.0165 vs 100%, $n$ = 9; *Figure 1E–G*), induced significant t-LTP when exogenous DA (20 µM) was bath-applied for 10–12 min from 2 min before and during the post-before-pre pairings in interleaved experiments (144 ± 12%; $t(5)$ = 3.7, p = 0.0148 vs 100%, $n$ = 6; *Figure 1E–G*). Therefore, in accordance with previous findings (*Zhang et al., 2009*; *Yang and Dani, 2014*), the presence of DA during the coordinated spiking activity widens the spike time interval for induction of t-LTP.

A crucial aspect of reinforcement learning models is the ability of the reinforcing signal (DA) to strengthen active synapses, even when it arrives after the activity (*Sutton and Barto, 1981*; *Izhikevich, 2007*). To test this hypothesis experimentally, we applied DA *after* the t-LTD induction protocol. Exogenous DA (100 µM) added to the perfusion system for 10–12 min starting within 1 min after the post-before-pre pairing protocol with $\Delta t$ = −20 ms converted t-LTD into t-LTP (169 ± 16%, $t(5)$ = 4.3, p = 0.0078 vs 100%, $n$ = 6; *Figure 2A,B*). This implies that DA can have a retroactive effect allowing negative spike pairings to induce t-LTP. The specificity of this DAergic conversion of STDP was assessed using DAR antagonists. Indeed, combined application of DAR antagonists,

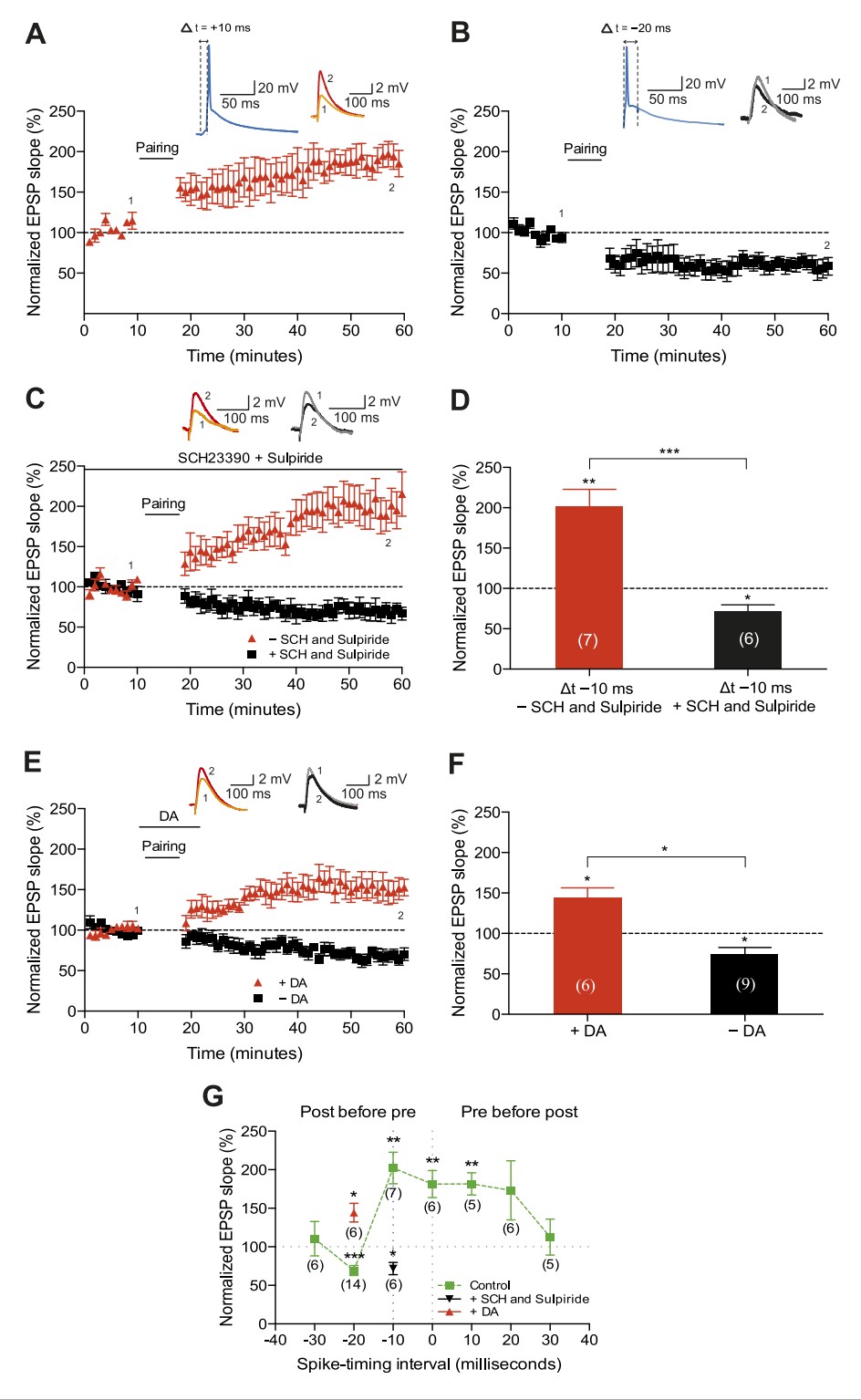

**Figure 1**. Dopamine widens the time window for induction of t-LTP. Example plots of normalized EPSP slopes demonstrating that (**A**) pre-before-post pairing protocol with Δt = +10 ms induced t-LTP, whereas (**B**) post-before-pre pairing protocol with Δt = −20 ms induced t-LTD. Insets, pairing protocols. Traces show an excitatory postsynaptic potential (EPSP) before (1) and 40 min after (2) pairing. (**C**) Endogenous DA widens the spike time window for induction of t-LTP. In the absence of DA receptor antagonists, the post-before-pre pairing protocol with Δt = −10 ms induced t-LTP (red), whereas application of SCH23390 and sulpiride at the start of the recordings

*Figure 1. continued on next page*

*Figure 1. Continued*
prevented this t-LTP and enabled t-LTD instead (black). Traces are presented as in **A**. (**D**) Summary of results. (**E**) Exogenous DA widens the spike time window for induction of t-LTP. In the presence of 20 µM DA, the post-before-pre pairing protocol with Δt = −20 ms induced t-LTP (red), whereas in control condition the same pairing protocol induced t-LTD (black). Traces are presented as in **A**. (**F**) Summary of results. (**G**) Summary of the spike timing-dependent plasticity (STDP) induction with various spike-timing intervals (Δt in ms) in control condition (green; data point at −20 ms represents data combined from **B** and **E**), in the presence of DA receptor antagonists (SCH23390 and sulpiride; black), or DA (red). Each data point is the group average EPSP slope percentage change from baseline. Error bars represent s.e.m. Significant difference (*p < 0.05, **p < 0.01, ***p < 0.001) compared with the baseline (one-sample two-tailed Student's *t*-test) or between the indicated two groups (paired two-tailed Student's *t*-test). The numbers of cells are shown in parentheses.
The following source data is available for figure 1:
**Source data 1**. Source data for *Figure 1*.

SCH23390 (10 µM) and sulpiride (50 µM), prevented DA-induced conversion of t-LTD into t-LTP, resulting in significant t-LTD instead (63 ± 12%, $t(5) = 3.0$, p = 0.0289 vs 100%, $n = 6$, Δt = −20 ms). Thus, the conversion of t-LTD into t-LTP was due to specific DAR activation. Importantly, when the test pathway was not stimulated following the pairing protocol until after DA washout (stimulation resumed 15 min after pairing), robust t-LTD was induced (61 ± 5%, $t(5) = 8.1$, p = 0.0005 vs 100%, $n = 6$; *Figure 2A,B*). The effect of DA was, therefore, activity dependent, demonstrating that the reinforcing signal is capable of acting specifically on the active inputs. To exclude the possibility that DA by itself could potentiate the test pathway, control experiments with ongoing synaptic stimulation over 60 min at 0.2 Hz, but without pairing with postsynaptic action potentials, were performed. Consistent with earlier reports (*Otmakhova and Lisman, 1999*), DA had no significant effect on the basal Schaffer-collateral transmission (110 ± 7%, $t(7) = 1.4$, p = 0.2169 vs 100%, $n = 8$; *Figure 2A,B*).

Subsequently, we wanted to determine whether the observed DA-induced conversion of t-LTD into t-LTP depends on the timing of DA application following the post-before-pre protocol (*Figure 3A–D*). We found that delayed application of DA (10 or 30 min after t-LTD pairing protocol) failed to convert t-LTD into t-LTP. Application of DA 10 min after the post-before-pre pairing caused a reversal of t-LTD back to baseline (94 ± 9%, $t(11) = 0.7$, p = 0.5230 vs 100%, $n = 12$; *Figure 3B,D*), whereas application of DA 30 min after the post-before-pre pairing failed to influence t-LTD altogether (59 ± 12%, $t(5) = 3.4$, p = 0.0200 vs 100%, $n = 6$; *Figure 3C,D*). Whilst it has previously been reported that DA applied after the induction of low frequency stimulation-induced LTD can reduce the magnitude of synaptic depression (*Mockett et al., 2007*), our data demonstrate, for the first time to our knowledge, that DA can change the polarity of STDP when acting within a short time window following the induction protocol.

Finally, we aimed to explore the possible mechanisms underlying the DA-induced conversion of t-LTD into t-LTP. Both hippocampal t-LTP and t-LTD (*Edelmann and Lessmann, 2011*; *Yang and Dani, 2014*), as well as the modulation of STDP by DA (*Zhang et al., 2009*), require functional NMDA receptors. We, therefore, asked whether the DA-induced conversion of t-LTD into t-LTP is also NMDA receptor dependent. Application of the NMDA receptor antagonist D-2-amino-5-phosphonopentanoic acid (D-AP5, 50 µM) after the post-before-pre pairing protocol did not by itself affect the development of t-LTD (57 ± 12%, $t(5) = 3.6$, p = 0.0156 vs 100%, $n = 6$, Δt = −20 ms; *Figure 4A,D*). Nevertheless, DA in the presence of D-AP5 reversed t-LTD back to baseline albeit failing to convert t-LTD into t-LTP (105 ± 12%, $t(5) = 0.4$, p = 0.6898 vs 100%, $n = 6$, Δt = −20 ms; *Figure 4A,D*). This suggests an important dissociation between two mechanisms involved in the DA-induced conversion of t-LTD into t-LTD, namely a reversal of synaptic depression (de-depression) and synaptic potentiation, one of which is NMDA receptor dependent.

To investigate the intracellular signaling mechanisms involved, we initially set out to establish the DAR subtype associated with the DA-induced conversion of t-LTD into t-LTP. Even though combined application of D1-like and D2-like receptor antagonist completely blocked the DA effect (*Figure 4—figure supplement 1A,D*), application of either D1-like or D2-like receptor antagonist alone only partially prevented the conversion of t-LTD into t-LTP (SCH 23390: 131 ± 16%; $t(6) = 1.4$, p = 0.0113 vs DA (t-LTP); $t(6) = 3.6$, p = 0.2277 vs control (t-LTD); $t(6) = 1.9$, p = 0.0994 vs 100%; $n = 7$.

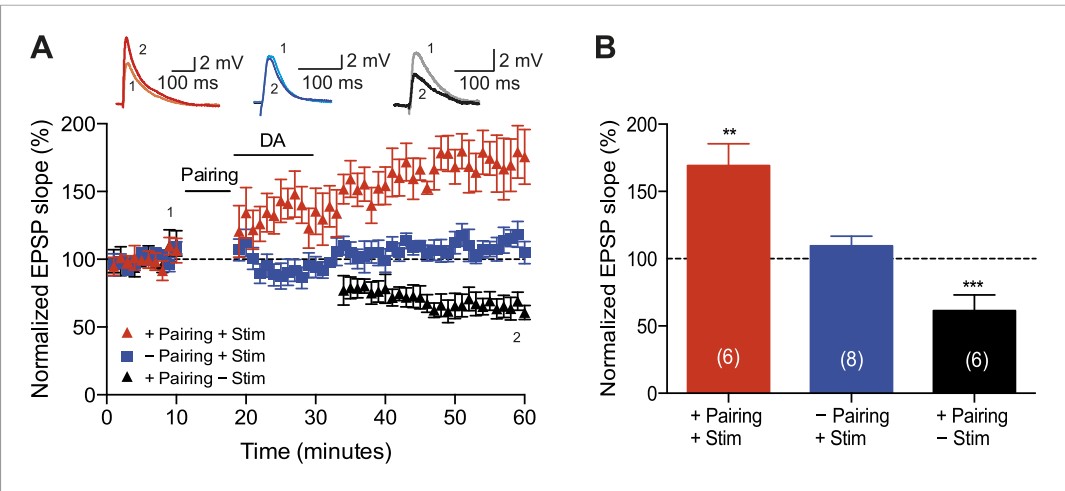

**Figure 2**. Dopamine retroactively converts t-LTD into t-LTP. (**A**) DA applied immediately after the pairing protocol with $\Delta t = -20$ ms converted t-LTD into t-LTP (+Pairing +Stim; red). If the pathway was not stimulated following the pairing protocol until after DA washout, t-LTD was induced (+Pairing −Stim; black). In the absence of the pairing protocol, DA had no effect on baseline EPSPs (−Pairing +Stim; blue). Traces show an EPSP before (1) and 40 min after (2) pairing. (**B**) Summary of results. Error bars represent s.e.m. Significant difference (\*\*p < 0.01, \*\*\*p < 0.001) compared with the baseline (one-sample two-tailed Student's t-test). The number of cells is shown in parentheses.

The following source data is available for figure 2:

**Source data 1**. Source data for *Figure 2*.

---

Sulpiride: $94 \pm 13\%$; $t(6) = 3.2$, p = 0.0228 vs DA (t-LTP); $t(6) = 1.5$, p = 0.1932 vs control (t-LTD); $t(6) = 0.5$, p = 0.6435 vs 100%; $n = 7$; *Figure 4—figure supplement 1B–D*). This suggests that both receptor subtypes might mediate the retroactive effect of DA on STDP. Given the astounding complexity of D2-like receptor pharmacology (*Neve et al., 2004*), we first wanted to evaluate the possible involvement of D1-like receptor-activated cAMP/PKA signaling cascade in the DA-induced conversion of t-LTD into t-LTP. D1/D5 receptor stimulation leads to the activation of adenylyl cyclase (AC) and subsequent increase in cyclic adenosine monophosphate (cAMP) and protein kinase A (PKA) activation (*Greengard et al., 1999*; *Neve et al., 2004*). We found that the general AC activator, forskolin (50 µM), applied for 10–12 min immediately after the pairing protocol with $\Delta t = -20$ ms resulted in robust conversion of t-LTD into t-LTP ($167 \pm 17\%$, $t(6) = 3.9$, p = 0.0078 vs 100%, $n = 7$; *Figure 4B,D*). Notably, forskolin (50 µM) applied in the absence of the pairing protocol had no significant effect on baseline EPSPs ($83 \pm 12\%$, $t(6) = 1.4$, p = 0.2046 vs 100%, $n = 7$; *Figure 4—figure supplement 2*). The forskolin-induced conversion of t-LTD into t-LTP was also NMDA receptor dependent since bath application of D-AP5 (50 µM) 1 min before forskolin treatment prevented synaptic potentiation from developing, although synaptic depression was reversed ($102 \pm 11\%$, $t(5) = 0.2$, p = 0.8616 vs 100%, $n = 6$; *Figure 4B,D*). This result suggests that the cAMP cascade works either upstream of or in parallel to NMDA receptor activation for the development of potentiation to occur. Downstream of cAMP, PKA is involved because the PKA inhibitor, H-89 (20 µM), blocked the DA-induced conversion of t-LTD into t-LTP, revealing significant t-LTD ($76 \pm 5\%$, $t(6) = 5.0$, p = 0.0025 vs 100%, $n = 7$; *Figure 4C,D*). Taken together, these results imply that the DA-induced conversion of t-LTD into t-LTP involves activation of the cAMP/PKA signaling cascade, which closely mimics the effects of DA (*Figure 4A* vs *Figure 4B*). Although, D2-like receptors are typically associated with the inhibition of AC (*Neve et al., 2004*), interestingly, there is also evidence that D2-like receptor stimulation can potentiate AC activity (*Glass and Felder, 1997*; *Watts and Neve, 1997*). Therefore, while the possibility that D2-like receptors contribute to the conversion of t-LTD into t-LTP via a different signaling cascade cannot be excluded, it is tempting to suggest that the DA-induced conversion of t-LTD into t-LTP is mediated primarily via the cAMP/PKA pathway. Hence, based on our

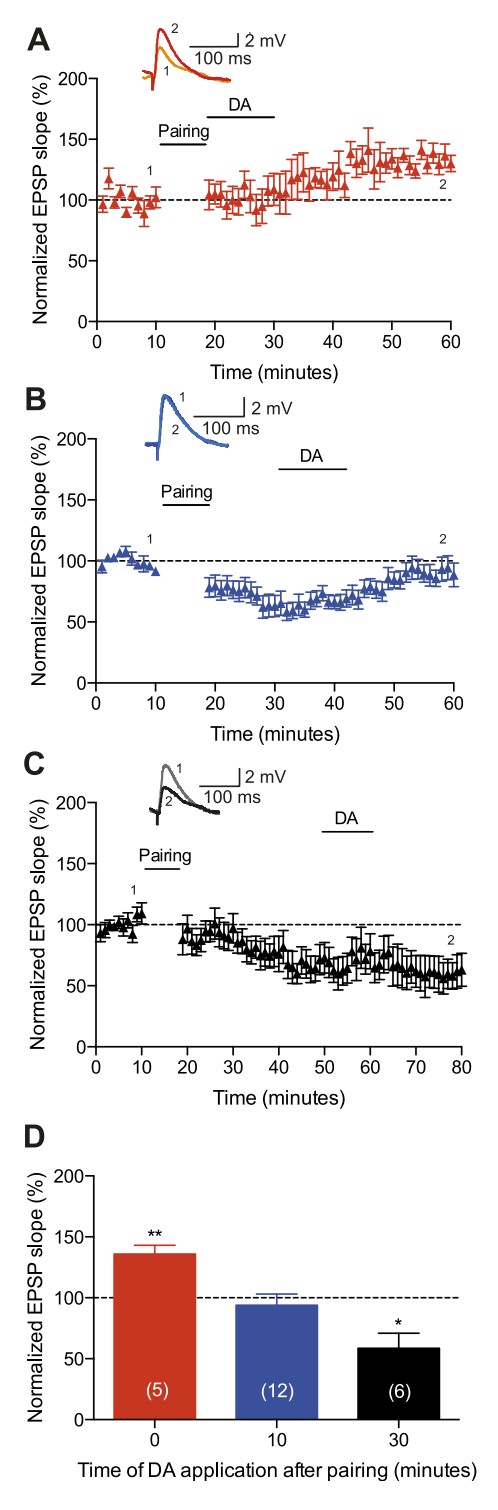

**Figure 3**. Time dependence of the DA-induced conversion of t-LTD into t-LTP. (**A**) DA applied immediately after the pairing protocol with Δt = −20 ms converted t-LTD into t-LTP, whereas delayed application of DA failed to convert t-LTD into t-LTP and either (**B**) resulted in a reversal of t-LTD back to baseline (10 min after pairing) or (**C**) failed to affect t-LTD

results, we propose that the stimulation of postsynaptic (*Figure 5ai*) or presynaptic (*Figure 5aii*) DARs activates the cAMP/PKA pathway, which—via two cellular mechanisms (de-depression and potentiation)—leads to the retroactive conversion of t-LTD into t-LTP.

The functional implications of our finding depend on the presynaptic source of DA in the hippocampus, which remains controversial because of the apparent discrepancy between DAergic terminals and receptors (*Scatton et al., 1980*; *Gasbarri et al., 1997*; *Lisman and Grace, 2005*). On one hand, it has been argued that noradrenergic terminals from the locus coeruleus, which mediates arousal and the optimization of behavioral performance (*Aston-Jones and Cohen, 2005*; *Chamberlain and Robbins, 2013*), may provide a major source of DA release (*Smith and Greene, 2012*). On the other hand, hippocampal pyramidal cell assemblies are directly affected by concurrent activity in midbrain DAergic neurons (*McNamara et al., 2014*), which have been linked to reward-seeking behavior (*Schultz et al., 1997*) and appetitive stimuli (*Mirenowicz and Schultz, 1996*; *Fiorillo et al., 2013*). This latter finding (*McNamara et al., 2014*) not only supports the hypothesis that hippocampal DA is relevant for reward processing, but is also consistent with our result that the activation of DAergic receptors *during* coordinated spiking activity changes the functional outcome of STDP (*Figure 1C–G*).

Previous studies examining reinforcement learning at the level of synaptic plasticity have showed that neuromodulators can affect timing-dependent plasticity in locust (*Cassenaer and Laurent, 2012*) and spine structural plasticity in striatal medium spiny neurons in mice (*Yagishita et al., 2014*) when acting within a delay time window of 1 s. While this narrow temporal detection window may be important in the striatum, it cannot account for the experimental evidence from behavioral studies of response acquisition with an extended reinforcement delay in rats and pigeons (*Lattal and Gleeson, 1990*; *Sutphin et al., 1998*), rhesus monkeys (*Galuska and Woods, 2005*), and humans (*Okouchi, 2009*). Meanwhile, our finding demonstrates that DA can modulate STDP in the CA1 with a reinforcement delay of at least 1 min (*Figure 5B*). Such extended reinforcement delay is likely to be particularly important in hippocampus-dependent learning during spatial exploration.

In conclusion, our work demonstrates a retroactive effect of DA on STDP—converting t-LTD

*Figure 3. Continued*

altogether (30 min after pairing). Traces show an EPSP before (1) and 40 min (**A**, **B**) or 60 min (**C**) after pairing (2). (**D**) Summary of results. Error bars represent s.e.m. Significant difference (*p < 0.05, **p < 0.01) compared with the baseline (one-sample two-tailed Student's *t*-test). The number of cells is shown in parentheses.

The following source data is available for figure 3:

**Source data 1**. Source data for *Figure 3*.

into t-LTP. This effect is mediated at least in part through the activation of the cAMP/PKA cascade and requires the activation of synaptic NMDA receptors. This in turn suggests that the conversion can only occur at synapses that are re-activated following the initial pairing event. Interestingly, it has been reported that hippocampal reactivation events (sharp wave ripples) increase in frequency following reward (*Singer and Frank, 2009*; *Atherton et al., 2015*). Thus, in behaving animals, the conditions for the conversion of depression into potentiation might occur during reward-related sharp wave ripple activity. Together, these findings support the concept of a slowly decaying synaptic *eligibility trace* that is committed to memory by the occurrence of reward and provide a possible mechanism for associating specific experiences with behaviorally distant, rewarding outcomes in animals (*Sutton and Barto, 1981*; *Suri and Schultz, 1999*; *Pan et al., 2005*; *Izhikevich, 2007*; *Harnett et al., 2009*), including humans (*Dunsmoor et al., 2015*).

## Materials and methods

### Animals
Wild-type mice (C57BL/6; postnatal days 12–18; from Harlan, Bicester, UK or Central Animal Facility, Physiological Laboratory, Cambridge University) of both sexes were housed on a 12-hr light/dark cycle at 19–23 °C, with water and food ad libitum. Experimental procedures and animal use were in accordance with the animal care guidelines of the UK Animals (Scientific Procedures) Act 1986 under personal and project licenses held by the authors. Caution was taken to minimize stress and the number of animals used in the experiments.

### Slice preparation
Mice were anesthetized with isoflurane and decapitated. The brain was rapidly removed, glued to the stage of a vibrating microtome (Leica VT 1200S, Leica Biosystems, Wetzlar, Germany) and immersed in ice-cold artificial cerebrospinal fluid (ACSF) containing the following (mM): 126 NaCl, 3 KCl, 26.4 $NaH_2CO_3$, 1.25 $NaH_2PO_4$, 2 $MgSO_4$, 2 $CaCl_2$, and 10 glucose. The ACSF solution, with pH adjusted to 7.2 and osmolarity to 270–290 mOsm $l^{-1}$, was continuously bubbled with carbogen gas (95% $O_2$/5% $CO_2$). The brain was sectioned into 350-μm-thick horizontal slices. The slices were incubated in ACSF at room temperature in a submerged-style storage chamber for at least 1 hr. For recordings (1–7 hr after slicing), individual slices were transferred to an immersion-type recording chamber, perfused with ACSF (2 ml $min^{-1}$) at 24–26 °C.

### Electrophysiology
#### Whole-cell recordings
Whole-cell patch-clamp recordings were performed on CA1 pyramidal neurons (located adjacent to the *stratum oriens*). For stimulation of Schaffer collaterals, a monopolar stimulation electrode was placed in the *stratum radiatum* of the CA1 subfield. The hippocampal subfields were visually identified using infrared differential interference contrast (DIC) microscopy. Patch pipettes (resistance: 4–8 MΩ) were made from borosilicate glass capillaries (0.68 mm inner diameter, 1.2 mm outer diameter), pulled using a P-97 Flaming/Brown micropipette puller (Sutter Instruments Co., Novato, California, USA). The internal solution of patch pipettes was (mM) 110 potassium gluconate, 4 NaCl, 40 HEPES, 2 ATP-Mg, 0.3 GTP (pH adjusted with 1 M KOH to 7.2, and osmolarity with $ddH_2O$ to 270 mOsm $l^{-1}$). The liquid junction potential was not corrected for. Cells were accepted for experiment only if the resting membrane potential at the start of the recording was between −55 and −70 mV. Membrane potential was held at −70 mV throughout further recording by direct current application via the recording electrode. At the beginning of each recording all cells were tested for regular spiking responses to positive current steps—characteristic of pyramidal neurons.

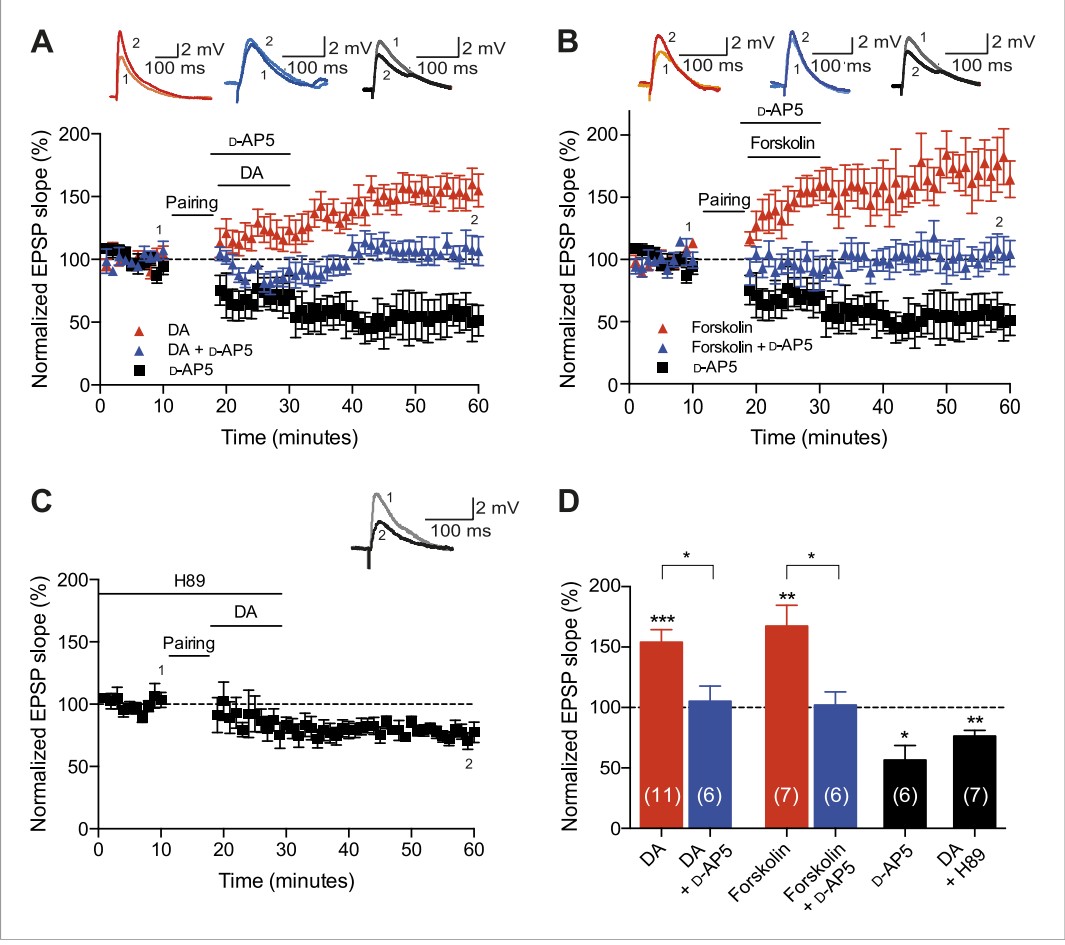

**Figure 4**. Cellular mechanisms involved in the DA-induced conversion of t-LTD into t-LTP. (**A**) DA applied immediately after the pairing protocol (Δt = −20 ms) converted t-LTD into t-LTP (red; data combined from *Figures 2A, 3A*). This effect requires NMDA receptors as application of D-AP5 1 min before the DA application partially blocked the conversion of t-LTD into t-LTP (blue), whereas application of D-AP5 alone failed to influence the development of t-LTD (black). Traces show an EPSP before (1) and 40 min after pairing (2). (**B**) The DA-induced conversion of t-LTD into t-LTP involves the activation of cAMP/PKA signaling cascade, which closely mimics the effect of DA. Forskolin, an AC activator, applied immediately after the pairing protocol (Δt = −20 ms) converted t-LTD into t-LTP (red). This effect requires NMDA receptors as application of D-AP5 1 min before forskolin application partially blocked the forskolin-induced conversion of t-LTD into t-LTP (blue), whereas application of D-AP5 alone failed to influence the development of t-LTD (black; data same as in **A**). (**C**) Downstream of cAMP, PKA is involved in the conversion of t-LTD into t-LTP as application of the PKA inhibitor, H-89, completely prevented the conversion of t-LTD into t-LTP. Traces are presented as in **A**. (**D**) Summary of results. Error bars represent s.e.m. Significant difference (*p < 0.05, **p < 0.01, ***p < 0.001) compared with the baseline (one-sample two-tailed Student's *t*-test) or between the indicated two groups (paired two-tailed Student's *t*-test). The number of cells is shown in parentheses.

The following source data and figure supplements are available for figure 4:

**Source data 1**. Source data for *Figure 4*.

**Source data 2**. Source data for *Figure 4—figure supplement 1*.

**Source data 3**. Source data for *Figure 4—figure supplement 2*.

**Figure supplement 1**. The retroactive conversion of t-LTP into t-LTD is due to specific DA receptor activation.

**Figure supplement 2**. Forskolin had no effect on baseline EPSPs.

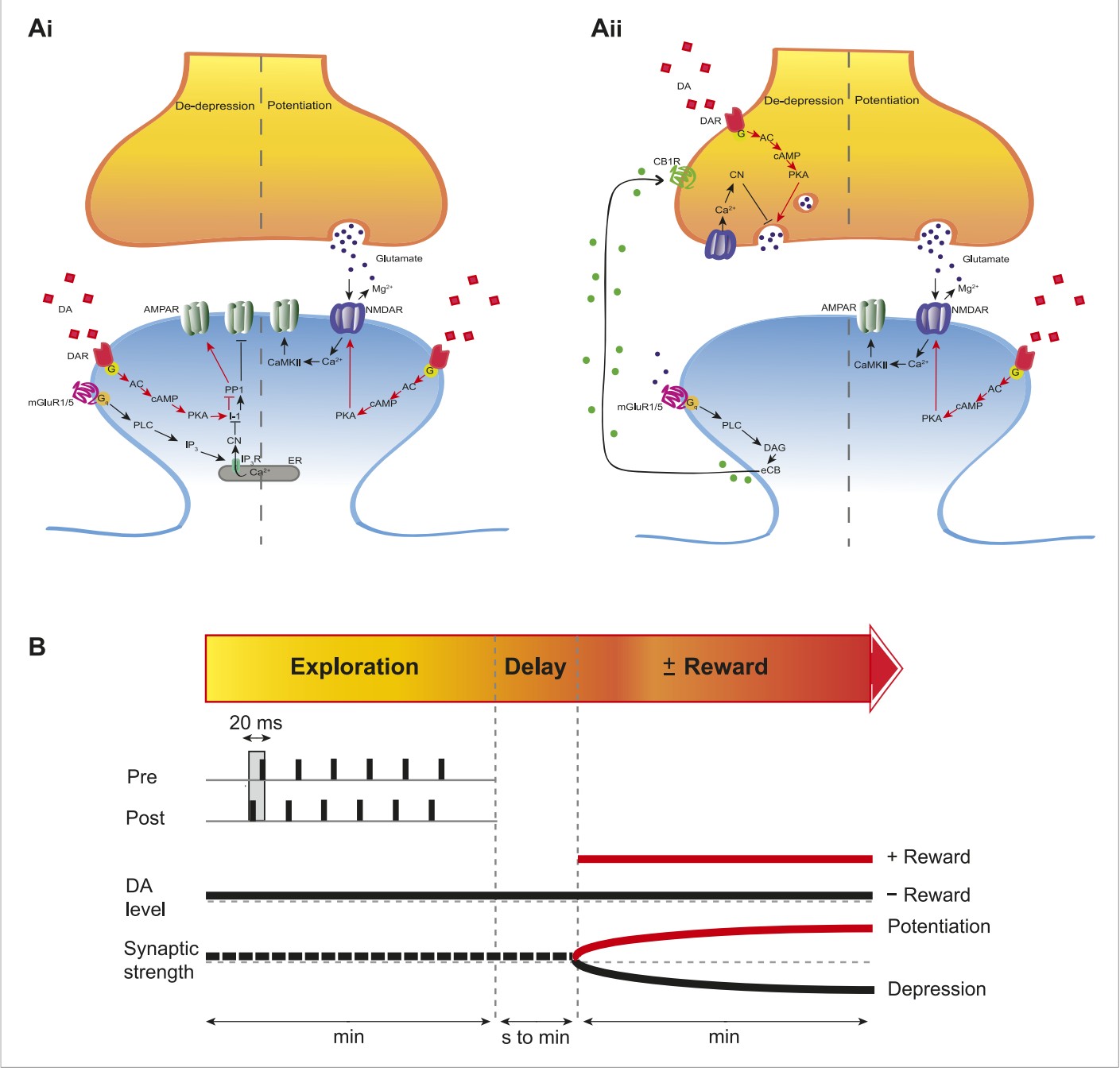

**Figure 5**. Proposed mechanisms underlying the DA-induced conversion of t-LTD into t-LTP. (**A**) Schematic diagram depicting core components of the proposed cellular mechanisms underlying the DA-induced conversion of t-LTD into t-LTP (de-depression and potentiation). (**Ai**) Model based on postsynaptic NMDAR-dependent potentiation (***Bi and Poo, 1998***; ***Caporale and Dan, 2008***; ***Zhang et al., 2009***; ***Edelmann and Lessmann, 2011***; ***Yang and Dani, 2014***) and metabotropic glutamate receptor-dependent (mGluRs) depression (***Otani and Connor, 1998***; ***Kemp and Bashir, 1999***; ***Huber et al., 2000***). De-depression (red, left): Activation of G protein-coupled D1/D5 receptors stimulates AC, increasing cAMP and activating PKA (***Greengard et al., 1999***; ***Neve et al., 2004***), which, via phosphorylation of I-1 (***Ingebritsen and Cohen, 1983***), reverses the PP1-induced dephosphorylation of synaptic AMPARs (***Lee et al., 2000***; ***Mockett et al., 2007***). Potentiation (red, right): PKA activation enhances NMDAR function (***Westphal et al., 1999***; ***Chen and Roche, 2007***). (**Aii**) Model based on presynaptic depression (***Bolshakov and Siegelbaum, 1994***; ***Siegel et al., 1994***; ***Oliet et al., 1997***; ***Charton et al., 1999***; ***Watabe et al., 2002***; ***Jourdain et al., 2007***). De-depression (red, left): Activation of presynaptic DA receptors stimulates AC, increasing cAMP and activating PKA (***Greengard et al., 1999***; ***Neve et al., 2004***), which reverses the calcineurin-dependent presynaptic depression. Potentiation (red, right): as in **Ai**. Arrow indicates activation/phosphorylation, blunt-ended line indicates inhibition/dephosphorylation. Abbreviations: AMPAR, AMPA-type glutamate receptor; NMDAR, NMDA-type glutamate receptor; mGluR1/5, group I metabotropic glutamate receptor; DAR,

*Figure 5. continued on next page*

*Figure 5. Continued*

dopamine receptor; AC, adenylate cyclase; cAMP, cyclic adenosine monophosphate; PKA, protein kinase A; I-1, inhibitor 1; PP1, protein phosphatase 1; PLC, phospholipase C; IP3, inositol 1,4,5-trisphosphate; ER, endoplasmic reticulum; DAG, diacylglycerol; eCB, endocannabinoid; CB1R, cannabinoid receptor type 1; CN, calcineurin; CaMKII, calcium-calmodulin-dependent protein kinase II. (**B**) Schematic diagram of synaptic and behavioral timescales in reward learning. During Exploration, the activity-dependent modification of synaptic strength due to spike timing-dependent plasticity (STDP) depends on the coordinated spiking between presynaptic and postsynaptic neurons on a millisecond time scale. Post-before-pre pairing leads to synaptic depression that develops gradually on a scale of minutes. When Reward, signaled via dopamine, follows Exploration with a Delay of seconds to minutes, synaptic depression is converted into potentiation.

## Stimulation protocol

EPSPs of amplitude between 3 and 8 mV were evoked at 0.2 Hz by adjusting the magnitude of direct current pulses (stimulus duration 50 μs, intensity 100 μA–1 mA). After a stable EPSP baseline period of at least 10 min, STDP was induced by repeated pairings of single presynaptic EPSP evoked by stimulation of Schaffer collaterals and single postsynaptic action potential elicited with the minimum somatic current pulse (1–1.8 nA, 3 ms) via the recording electrode. Pairings were repeated 100 times at 0.2 Hz. Spike-timing intervals (Δt in ms) were measured between the onset of the EPSP and the onset of the action potential. The EPSPs were monitored for at least 40 min after the end of the pairing protocol. Presynaptic stimulation frequency remained constant throughout the experiment.

## Data acquisition and data analysis

Voltage signals were low-pass filtered at 2 kHz using an Axon Multiclamp 700B amplifier (Molecular Devices, Sunnyvale, California, USA). Data were acquired at 5 kHz via an ITC18 interface board (Instrutech, Port Washington, New York, USA), transmitting to a Dell computer running the Igor Pro software (WaveMetrics, Lake Oswego, Oregon, USA). All experiments were carried out in the current clamp ('bridge') mode. Series resistance was monitored (10–15 MΩ) and compensated for by adjusting the bridge balance. Data were discarded if series resistance changed by more than 30%. Data were analyzed using Igor Pro. EPSP slopes were measured on the rising phase of the EPSP as a linear fit between the time points corresponding to 25–30% and 70–75% of the peak amplitude. For statistical analysis, the mean EPSP slope per minute of the recording was calculated from 12 consecutive sweeps and normalized to the baseline. Normalized ESPS slopes from the last 5 min of the baseline (immediately before pairing) and from the last 5 min of the recording (35–40 min or 55–60 min after pairing) were averaged. The magnitude of plasticity, as an indicator of synaptic change, was defined as the average EPSP slope after pairing expressed as a percentage of the average EPSP slope during baseline.

## Drugs

The following drugs were used: dopamine hydrochloride 20 μM, forskolin 50 μM, D-AP5 50 μM, SCH23390 hydrochloride 10 μM, sulpiride 50 μM, H-89 20 μM. All drugs (purchased from Sigma–Aldrich, Dorset, United Kingdom; Tocris Bioscience, Bristol, United Kindgom; or Abcam, Cambridge, United Kingdom) were bath-applied through the perfusion system by dilution of concentrated stock solutions (prepared in water or DMSO) in ACSF.

## Statistical analysis

Statistical comparisons were made using one-sample two-tailed or paired two-tailed Student's *t*-test, with a significance level of α = 0.05. Data are presented as mean ± s.e.m. Significance levels are indicated by *$p < 0.05$, **$p < 0.01$, ***$p < 0.001$.

## Acknowledgements

This research was supported by a studentship from the Medical Research Council (UK) to ZB, the School of Biological Sciences at the University of Cambridge, and the Wellcome Trust (WS).

## Additional information

### Competing interests
WS: Reviewing editor, *eLife*. The other authors declare that no competing interests exist.

### Funding

| Funder | Grant reference | Author |
|---|---|---|
| Medical Research Council (MRC) | Graduate studentship | Zuzanna Brzosko |
| University of Cambridge | Student research support fund | Zuzanna Brzosko, Wolfram Schultz, Ole Paulsen |

The funders had no role in study design, data collection and interpretation, or the decision to submit the work for publication.

### Author contributions
ZB, Conception and design, Acquisition of data, Analysis and interpretation of data, Drafting or revising the article; WS, OP, Conception and design, Drafting or revising the article

### Author ORCIDs
Ole Paulsen, http://orcid.org/0000-0002-2258-5455

### Ethics
Animal experimentation: Experimental procedures and animal use were in accordance with the animal care guidelines of the UK Animals (Scientific Procedures) Act 1986 under Home Office personal license PIL- ICB486697 and project license PPL80/2440 held by the authors. Caution was taken to minimize stress and the number of animals used in experiments.

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
