## [Decision Letter]

Thank you for submitting your work entitled “Retroactive effect of dopamine on spike timing-dependent plasticity” for further consideration at *eLife*. Your article has been favorably evaluated by Eve Marder (Senior editor) and three reviewers, one of whom is a member of our Board of Reviewing Editors. One of the reviewers, Guoqiang Bi, has agreed to reveal his identity.

All three reviewers viewed the study as valuable for the readership of *eLife*. The main observation that spike-time dependent LTD can be converted into LTP upon DA release 10 min after the initial LTP-pairing protocol, was viewed as interesting. However, major criticism was formulated in relation to the missing explanation on how the timing of bath-applied DA may relate to reward-based conditioning. The reviewers asked for diagrams describing the timing issues, in particular the possible linkages between the cAMP pathway and its relation to Ca^2+^ signaling caused by action potentials and NMDAR activation. A stronger discussion of the data with existing literature is necessary. Moreover, one reviewer asked for additional experiments in which SCH23390 and sulpiride are separately applied. The reviewers were also curious for your thoughts about how the possible cAMP/PKA-mediated mechanism could be linked to Ca^2+^ signals, NMDA receptor activation and finally the switch from LTD to LTP. The specific comments of the reviewers are attached below.

Reviewer #1:

The study by the Paulsen's group shows nicely that spike-time dependent LTD can be converted into LTP upon DA release 10 min after the initial LTP-pairing protocol. They propose and show good experimental evidence for their hypothesis that this is caused by activation of the cAMP/PKA pathway plus activation of NMDARs. The data fit very well with a recent study from McNamara et al. (Nature Neuroscience, 2014) showing that DA improves reactivation of cell assemblies in CA1 and memory persistence. I have mild concerns regarding the precise timing of bath-applied drugs in the bath. How fast does the drug get to the DA receptors located at the recorded cell? This is important in defining the time window during which the DA effect can take place and induce the observed conversion of synaptic plasticity. The difference in the finding presented here to the one by the McNamara group should be emphasized.

Reviewer #2:

This study by Brzosko et al. addressed how dopamine (DA) retroactively regulates spike timing-dependent plasticity (STDP) in order to account for the distal reward problem, where rewards promote learning of experiences preceding rewards. The main result is that DA applied after the delivery of the post-before-pre pairing protocol converted LTD into LTP.

1) DA was bath applied for 10-12 min starting within 1 min after the delivery of the post-before-pre pairing protocol, which took ∼8 min. It is not clear how this relates to behaviorally relevant timing during reward-based conditioning, as described in [40], [45], and [37]. Furthermore, how does the millisecond order timing dependence in STDP relate to the second order timing dependence during reward-based learning (for example, see Drew and Abbott, PNAS, 2006, “Extending the effects of spike-timing-dependent plasticity to behavioral timescales”)? A schematic diagram on this timing issue (from millisecond to second to min in the current study) should be provided to account for the behavioral relevance.

2) In terms of the cellular mechanism, it is not clear how DA-induced activation of cAMP/PKA pathway relates to Ca signals caused by spikes and NMDA receptor activation. A schematic diagram describing the possible linkage should be provided with the timing diagram mentioned above. The authors may want to compare their data with a recent paper that addressed this issue on DA/spike timing dependent plasticity of dendritic spines (Yagishita et al., Science, 2015, A critical time window for dopamine actions on the structural plasticity of dendritic spines).

3) In the bar graphs, it is not clear what the significance mark on each bar means. I do understand the significance marks between bars.

4) It should be stated if any of the treatments (plasticity induction, DA/forskolin application, etc.) affected the amount of current necessary to hold the membrane potential at -70 mV, preferably with actual data.

5) According to the description in the Results section, LTP induced with -10 ms post-before-pre timing was “surprising”. This effect is ascribed to the release of endogenous DA by the stimulating electrode. It should be discussed in further detail how this observation compares to previous studies in brain slices and cultures.

Reviewer #3:

In this manuscript, the authors investigated the temporal specificity of dopaminergic modulation of STDP. They found that dopamine application within 10 minutes after STDP pairing could retroactively convert t-LTD into t-LTP. This conversion is apparently mediated by cAMP/PKA signaling, and requires synaptic activation as well as functional NMDA receptors. Together, the results provide an elegant cellular mechanism for the delayed effect of dopamine in reinforcement learning.

Overall I think the experiments were well designed and carried out. The results are important and intriguing, and should be of interests for both cellular and systems neuroscientists. I do have several questions:

1) Throughout the study, the authors used combined SCH23390+sulpiride to block both D1/D5-like receptors and D2-like receptors. Can the authors test differential roles of the two receptor types in STDP conversion by using the two antagonists separately, as has been done in previous studies?

2) Regarding the cellular signaling mechanisms, the authors argue that since AP-5 blocks the effect of DA and forskolin in t-LTD/LTP conversion, the cAMP/PKA signaling works either upstream of, or in parallel to NMDA receptor activation. Can the authors further allude to a potential mechanism(s) how such signaling interaction could happen?

---

## [Author Response]

Reviewer #1:

The study by the Paulsen's group shows nicely that spike-time dependent LTD can be converted into LTP upon DA release 10 min after the initial LTP-pairing protocol. They propose and show good experimental evidence for their hypothesis that this is caused by activation of the cAMP/PKA pathway plus activation of NMDARs. The data fit very well with a recent study from McNamara et al. (Nature Neuroscience, 2014) showing that DA improves reactivation of cell assemblies in CA1 and memory persistence. I have mild concerns regarding the precise timing of bath-applied drugs in the bath. How fast does the drug get to the DA receptors located at the recorded cell? This is important in defining the time window during which the DA effect can take place and induce the observed conversion of synaptic plasticity. The difference in the finding presented here to the one by the McNamara group should be emphasized.

The study by [29] shows that optogenetic activation of midbrain dopaminergic neurons *during* spatial exploration and learning enhances reactivation of newly encoded hippocampal representations and improves memory performance. This is consistent with our results demonstrating that the activation of DAergic receptors *during* coordinated spiking activity changes the functional outcome of STDP such that negative time pairings induce potentiation and not depression (Figure 1). Please note, however, that the main finding of our study is significantly extending that of McNamara et al. in showing that dopamine can convert hippocampal timing-dependent synaptic depression into potentiation even when applied *after* the pairing event (as relevant for addressing the distal reward problem in the field of reinforcement learning). This effect was seen when dopamine was applied within one minute after the end of the pairing event. This has now been emphasized in the Results and Discussion section.

Reviewer #2:

*1) DA was bath applied for 10-12 min starting within 1 min after the delivery of the post-before-pre pairing protocol, which took ∼8 min. It is not clear how this relates to behaviorally relevant timing during reward-based conditioning, as described in*
[40]*,*
[45]*, and*
[37]*.*

Our data show that DA can modulate STDP in the CA1 with a reinforcement delay of at least one minute (Figure 5). This extended reinforcement delay is likely to be important in hippocampus-dependent memory tasks during spatial exploration.

Although behavioral evidence from hippocampus-based tasks is still lacking, our results are consistent with behavioral conditioning as demonstrated in multiple studies of response acquisition with delayed reinforcement in other brain areas: in rats and pigeons (25; 46), rhesus monkeys (14) and humans (33). This has now been emphasized in the manuscript (Results and Discussion section).

Please note that the duration of DA application (i.e. 10-12 minutes) mirrors the slow time profile of the gradually developing synaptic depression, where the synaptic weights reach their maximum depressed state after ∼10-15 minutes.

Furthermore, how does the millisecond order timing dependence in STDP relate to the second order timing dependence during reward-based learning (for example, see Drew and Abbott, PNAS, 2006, “Extending the effects of spike-timing-dependent plasticity to behavioral timescales”)? A schematic diagram on this timing issue (from millisecond to second to min in the current study) should be provided to account for the behavioral relevance.

We would like to thank for the suggestion of incorporating such a schematic, which is now presented in Figure 5.

2) In terms of the cellular mechanism, it is not clear how DA-induced activation of cAMP/PKA pathway relates to Ca signals caused by spikes and NMDA receptor activation. A schematic diagram describing the possible linkage should be provided with the timing diagram mentioned above.

Again, we would like to thank you for this suggestion, and we have incorporated a figure to illustrate the links between the cAMP/PKA signaling cascade and the consensus signaling events underlying the induction of LTP and LTD at hippocampal synapses (Figure 5).

The authors may want to compare their data with a recent paper that addressed this issue on DA/spike timing dependent plasticity of dendritic spines (Yagishita et al., Science, 2015, A critical time window for dopamine actions on the structural plasticity of dendritic spines).

To address this request, we have incorporated the following paragraph to the Results and Discussion: “Previous studies examining reinforcement learning at the level of synaptic plasticity have showed that neuromodulators can affect timing-dependent plasticity in locust […]. Such extended reinforcement delay is likely to be particularly important in hippocampus-dependent learning during spatial exploration.”

We would like to emphasize that the timing rules for reinforcement are likely to be variable across different systems, ranging from 0.5-1.0 sec time delays for dopamine learning (18) and fast striatal processing (51), via 1-40 sec for more complex behavioral tasks (25; 46; 14; 33) and aversive stimuli (Dickinson, Contemporary animal learning theory. Cambridge, UK: Cambridge University Press, 1980), to delays in the order of at least one minute for hippocampus-dependent learning as observed in our study.

3) In the bar graphs, it is not clear what the significance mark on each bar means. I do understand the significance marks between bars.

The significance mark on each bar indicates significant difference (**P* < 0.05, ***P* < 0.01, ****P* < 0.001) compared with the baseline (one-sample two-tailed Student’s *t*-test) or between the indicated two groups (paired two-tailed Student’s *t*-test). This has now been explicitly stated in each figure legend.

*4) It should be stated if any of the treatments (plasticity induction, DA/forskolin application, etc.) affected the amount of current necessary to hold the membrane potential at -70 mV, preferably with actual data.*

Author response image 1.Example plot showing changes in the holding current over time for post-before-pre pairing protocol with Δt = −20 ms in control t-LTD (n = 9) and with DA added after pairing (n = 11). Error bars represent s.e.m.**DOI:**
http://dx.doi.org/10.7554/eLife.09685.018

*While there was a small difference in holding current between Control t-LTD and DA condition*, a similar difference was not seen with Forskolin, suggesting that the change in holding current is not likely to be related to the effect on synaptic efficacy. We believe that this figure does not contribute to explaining our finding and, if possible, we would prefer not to include it in the manuscript.

5) According to the description in the Results section, LTP induced with -10 ms post-before-pre timing was “surprising”. This effect is ascribed to the release of endogenous DA by the stimulating electrode. It should be discussed in further detail how this observation compares to previous studies in brain slices and cultures.

In view of the conventional presentation of STDP, we found this finding surprising. We now explicitly state that “This conflicts with previous reports from hippocampal cultures (3; 53) and acute slices (11; 52), where post-before-pre pairing protocols never elicited synaptic potentiation in baseline condition.”

However, it has been shown earlier that DA widens the time window for the induction of t-LTP (53; 52), and this has now been discussed in the same paragraph.

Reviewer #3:

1) Throughout the study, the authors used combined SCH23390+sulpiride to block both D1/D5-like receptors and D2-like receptors. Can the authors test differential roles of the two receptor types in STDP conversion by using the two antagonists separately, as has been done in previous studies?

In response to this comment, we have carried out a new set of experiments, applying D1-like or D2-like receptor antagonist separately. Even though combined application of D1-like and D2-like receptor antagonist completely blocked the DA effect (Figure 4—figure supplement 1), application of either D1-like or D2-like receptor antagonist alone only partially prevented the conversion of t-LTD into t-LTP (SCH 23390: 131 ± 16%; *t*(6) = 1.4, *P* = 0.0113 vs. DA (t-LTP); *t*(6) = 3.6, *P* = 0.2277 vs. control (t-LTD); *t*(6) = 1.9, *P* = 0.0994 vs. 100%; *n* = 7. Sulpiride: 94 ± 13%; *t*(6) = 3.2, *P* = 0.0228 vs. DA (t-LTP); *t*(6) = 1.5, *P* = 0.1932 vs. control (t-LTD); *t*(6) = 0.5, *P* = 0.6435 vs. 100%; *n* = 7; Figure 4—figure supplement 1). This suggests that both receptor sub-types might mediate the retroactive effect of DA on STDP. This is now reported in the text (in the sixth paragraph of the Results and Discussion section), and presented in Figure 4—figure supplement 1.

We further discuss the implications of this in the same paragraph as follows: “D1/D5-receptor stimulation leads to the activation of adenylyl cyclase (AC) and subsequent increase in cyclic adenosine monophosphate (cAMP) […] to the retroactive conversion of t-LTD into t-LTP.”

2) Regarding the cellular signaling mechanisms, the authors argues that since AP-5 blocks the effect of DA and forskolin in t-LTD/LTP conversion, the cAMP/PKA signaling works either upstream of, or in parallel to NMDA receptor activation. Can the authors further allude to a potential mechanism(s) how such signaling interaction could happen?

Please see the response to Reviewer 2, point 1 above.

Additional changes to the manuscript:

To clarify the presentation in accordance to the reviewers’ comments, we have also updated Figure 1 and Figure 4, and expanded and moved Figure 2–figure supplement 1 to Figure 4—figure supplement 1.